# The Usefulness of Serological Inflammatory Markers in Patients with Rotator Cuff Disease—A Systematic Review

**DOI:** 10.3390/medicina58020301

**Published:** 2022-02-16

**Authors:** Chi Ngai Lo, Bernard Pui Lam Leung, Shirley Pui Ching Ngai

**Affiliations:** 1Family Care Physiotherapy Clinic, 612 Clementi West St. 1, Singapore 120612, Singapore; 2Health and Social Sciences Cluster, Singapore Institute of Technology, 10 Dover Drive, Singapore 138683, Singapore; bernard.leung@singaporetech.edu.sg; 3Department of Rehabilitation Sciences, The Hong Kong Polytechnic University, Hong Kong SAR, China; shirley.ngai@polyu.edu.hk

**Keywords:** rotator cuff disease, biomarkers, blood test, systematic review

## Abstract

*Background and Objectives*: Rotator cuff disease (RCD) is a prominent musculoskeletal pain condition that spans a variety of pathologies. The etiology and precise diagnostic criteria of this condition remain unclear. The current practice of investigating the biochemical status of RCD is by conducting biopsy studies but their invasiveness is a major limitation. Recent biochemical studies on RCD demonstrate the potential application of serological tests for evaluating the disease which may benefit future clinical applications and research. This systematic review is to summarize the results of available studies on serological biochemical investigations in patients with RCD. *Methods*: An electronic search on databases PubMed and Virtual Health Library was conducted from inception to 1 September 2021. The inclusion criteria were case-control, cross-sectional, and cohort studies with serological biochemical investigations on humans with RCD. Methodological quality was assessed using the Study Quality Assessment Tool for Observational Cohort and Cross-sectional studies from the National Heart, Lung, and Blood Institute. *Results:* A total of 6008 records were found in the databases; of these, 163 full-text studies were checked for inclusion and exclusion criteria. Nine eligible studies involving 984 subjects with RCD emerged from this systematic review. The quality of the studies found ranged from poor to moderate. In summarizing all the studies, several fatty acids, nonprotein nitrogen, interleukin-1 β, interleukin-8, and vascular endothelial growth factor were found to be significantly higher in blood samples of patients with RCD than with control group patients, while Omega-3 Intex, vitamin B12, vitamin D, phosphorus, interleukin-10, and angiogenin were observed to be significantly lower. *Conclusions*: This is the first systematic review to summarize current serological studies in patients with RCD. Results of the studies reflect several systemic physiological changes in patients with RCD, which may prove helpful to better understand the complex pathology of RCD. In addition, the results also indicate the possibility of using serological tests in order to evaluate RCD; however, further longitudinal studies are required.

## 1. Introduction

Rotator cuff disease (RCD) spans a variety of pathologies including rotator cuff (RC) tendon impingement in the subacromial space, partial tendon tears, full-thickness tears, and cuff tear arthropathy [1]. RCD is a prominent condition in musculoskeletal pain, with a prevalence reported to be between 18% and 31% in the general population, and in over 66.7% of people throughout their lifetimes [2]. Moreover, the incidence rate of surgical intervention for RCD has been significantly increasing in the US and Australia [3,4].

The etiology and precise diagnostic criteria of this condition are still unclear. Recently, multiple studies have observed significantly elevated levels of pro-inflammatory mediators in biopsies of rotator cuff tendons in patients with RCD [5,6], indicating that inflammatory processes may play a significant role in the development of disease [7]. These findings are helpful to explore underlying pathologies and potential diagnostic tools. Recent research evidence demonstrates an association between RCD and metabolic syndrome [8]. However, the invasiveness of biopsy studies in RCD remains a major limitation as biopsies hinder the availability of repetitive measures in longitudinal biochemical studies and in clinical disease monitoring.

Serological investigation is a more convenient, lower cost, less time-consuming, and less invasive means to evaluate systemic bio-physiological changes. Increasing evidence supports serological investigation for RCD, and raises the prospect of using serological biomarkers in monitoring the responsiveness of RCD to current treatments. At present, it remains unclear if any specific biomarker can reflect disease status, severity, and clinical implications. The aim of this systematic review is to summarize recent findings from available serological biochemical studies in patients with RCD compared to people without shoulder pathologies.

## 2. Materials and Methods

### 2.1. Study Design and Registration

This systematic review was conducted following procedures in the Cochrane Handbook [9], and the review is reported according to the Preferred Reporting Items for Systematic Review and Meta-Analysis (PRISMA) guidelines [10]. The protocol was registered on the International Prospective Register of Systematic Reviews (PROSPERO), CRD42021224718.

### 2.2. Search Strategy

To summarize and update available reports on physiological changes in patients with RCD, a literature search using electronic search engines PubMed (including MEDLINE and PubMed Central) and Virtual Health Library (including MEDLINE, Latin America and Caribbean Literature in Health Sciences LILACS, Índice Bibliográfico Espanhol de Ciências da Saúde IBECS, and BENF Nursing) was conducted from inception to 1 September 2021. The language of articles chosen was limited to English only. The keywords and search strategy used are detailed in Section A.1 and Section A.2. The reference lists of the included articles in addition to Google Scholar were checked for additional eligible articles.

### 2.3. Eligibility Criteria

Student Design—inclusion criteria were case-control studies, cross-sectional studies, or cohort studies.Population—subjects of the studies comprised adult (>18 years old) humans with RCD, including diagnosis of RC tendinopathy, shoulder impingement, subacromial bursitis, and RC tears. The diagnoses of RCD were confirmed by radiological or surgical examinations. Study populations involving other shoulder pathologies such as fractures, frozen shoulders, and shoulder joint dislocations were excluded. Subjects with other systemic conditions or comorbidities which could affect the outcomes were excluded. In addition, the studies were accompanied by suitable control groups without pathologies in the shoulders or without systemic conditions such as inflammatory joint disease and without immunological or neoplastic disorders. All animal studies were excluded.Outcomes—the studies needed to have outcome measures that could objectively quantify physiological biomarkers in blood samples of the subjects. Studies that lacked statistical analyses to compare their outcomes with were excluded.

### 2.4. Study Selection

Search records were imported into the referencing website EndNote Web (https://access.clarivate.com, accessed on 6 September 2021) for duplication elimination. Based on the inclusion criteria, abstracts and titles were primarily screened by two independent reviewers (CL and BL), followed by screening of the full-text articles. A third reviewer (SN) resolved disagreements. Reviewer CL is a physiotherapist with more than 15 years of clinical experience, a masters degree with research training, and more than 10 years of experience in musculoskeletal clinical research. Reviewer BL is a researcher with a PhD degree in immunology and more than 25 years of research experience in biomarkers and inflammatory diseases. Reviewer SN is a researcher with a PhD degree in physiotherapy with specialized training in epidemiology and biostatistics, plus more than 15 years of research experience in exercise physiology. Manual searching of included studies’ reference lists was performed to identify additional eligible studies. Meta-analysis was planned to determine if selected articles have comparable outcome measures.

### 2.5. Quality and Risk of Bias Assessment

Selected articles were appraised on quality using the Study Quality Assessment Tool for Observational Cohort and Cross-sectional studies from the National Heart, Lung, and Blood Institute (https://www.nhlbi.nih.gov/health-topics/study-quality-assessment-tools, (accessed on 30 September 2021). This assessment tool consists of 14 questions to evaluate each study’s internal validity and risks of selection bias, information bias, measurement bias, or confounding bias. Quality and risk of bias assessment were conducted by two reviewers (CL and BL) individually. Any inconsistent results were discussed and resolved together with the third reviewer (SN).

### 2.6. Data Extraction

Data were extracted from selected papers by one reviewer (CL) and checked by a second reviewer (SN) using a piloted table designed by the authors. The data included study design, demographics of the subjects, sample size, diagnosis for RCD, types of serological measures, results, and statistical analysis. If data in a selected study were missing or not clear, its authors were contacted to request details. The primary focus of the results was on identifying differences in the average levels of serological biomarkers between study groups and control groups in addition to statistical significances, if any.

## 3. Results

The process flow of initial search results, screening, inclusion, and exclusion of full texts is illustrated in Figure 1. A total of 6008 records were found in the selected databases. After the removal of duplicates, 3470 records were screened using an automated filter for English, human subjects, study designs, along with reviewers screening for titles and abstracts. A total of 163 studies were checked through full-text screening; of those, 152 studies were excluded since their study designs or study populations did not match the selection criteria. Two more studies [11,12] were excluded during data extraction since their outcome measures are genetic factors rather than biochemical substances. Ultimately nine studies were selected from this systematic review.

Figure 1 Flowchart of the literature search.

### 3.1. Study Characteristics

Characteristics of the included studies are presented in Table 1. These articles comprised eight case-control studies and one cohort study. Most of the population in the experimental group were patients with RC tears confirmed by MRI or surgical investigation and clinical tests but Sayitskaya et al.’s (2011) study also included patients with RC inflammation and tendon degeneration [13]. From the included studies, a total of 984 patients with RCD comprised the experimental group. The control group’s subjects were similar in gender ratio and age with those of the experimental group but without evidence of shoulder pathologies or conditions that could have affected the serological tests. Serological assessments were conducted through plasma or serum samples in order to measure levels of matrix metalloproteinase (MMP), tissue inhibitors of metalloproteinases (TIMPs), fatty acids, Omega-3 index, cholesterols, fasting plasma glucose, triglyceride, fibrinogen, nonprotein nitrogen (NPN), creatinine, interleukins (ILs), vascular endothelial growth factor (VEGF), angiogenin (ANG), vitamin D, vitamin B12, and other substances.

The levels of fatty acids, nonprotein nitrogen, interleukin-1β, IL-8, and VEGF were significantly higher in the blood samples of patients with RCD than those in the control group. In contrast, the levels of Omega-3 Index, vitamin B12, vitamin D, phosphorus, IL-10, and ANG were significantly lower. Although Hallgren et al. (2012) found elevated plasma levels of TIMP-1 in their study group, the result is uncertain because of differences reported between the investigations by Luminex and ELISA methods [14]. Other biomarkers were reported as either not statistically significant or clinically different between experimental and control groups.

### 3.2. Quality and Risk of Bias of Selected Articles

According to the Levels of Evidence for Prognostic Studies, the nine included studies were level III evidence [15]. Assessment of risk of bias and methodologic quality is presented in Table 2. Possible risk of bias could be present as most selected studies do not demonstrate sample size justification. All the studies are cross-sectional and the timeframe is insufficient to support an association between a biochemical substance and RCD. It is possible for the studies to have repetitive measures and longer follow-up periods yet the studies only feature single measurements of serological tests. The selected studies fulfil five to nine out of the fourteen items listed on the assessment tool, with item 8 not being applicable. The quality of the studies is considered poor to fair.

## 4. Discussion

Results of RCD immunologic biomarker studies through biopsy show significant increases in pro-inflammatory markers cyclooxygenase COX-1 and COX-2, TNFα, IL-1β, IL-6, hypoxia-inducible factors (HIFs), VEGF, and degenerative enzymes MMP-1, MMP-9, and MMP-13 in patients with RCD [16,17,18,19,20,21,22,23,24,25,26,27,28,29,30,31,32,33,34,35,36,37,38,39,40,41,42,43,44]. Compared with current available serological studies, significantly higher levels of IL-1β and VEGF [13] were also found in blood samples of the patients but not MMP-1 and/or MMP-9 [14]. The remainder of the pro-inflammatory markers are pending further investigation.

Importantly, Savitskaya et al. (2011) observed that changes in serum levels of IL-1β, IL-8, VEGF, ANG, and IL-10 are statistically significant and specific in distinguishing between patients with RCD and control subjects [13]. As stated, levels of IL-1β and VEGF are demonstrated to be significantly higher in patients with RCD through both biopsy and serum samples; this may indicate that the inflammatory process is involved in RCD locally and systemically. Interleukin-1β is a cytokine serving as an inflammatory mediator which induces the expressions of MMP-1, MMP-3, MMP-13, IL-1, IL-6, and COX-2 in tendon cells. It functions for the modulation of immune responses and tissue remodelling [41,45]. Significant correlations were also computed between overexpression of VEGF and clinical presentations of advanced stages of RCD (r = 0.75; *p <* 0.01), average microvascular density (r = 0.68, *p* < 0.01), and visual analog scores (r = 0.75, *p <* 0.01). VEGF is a pro-inflammatory substance and signal protein that is produced by endothelial and mesothelial cells which can be found in blood vessels and internal organs [46,47,48]. It plays an important role in angiogenesis for migration and mitosis of endothelial cells, increasing matrix metalloproteinase activities, stimulating vessels to invade hyper-vascularized tissue, and producing fenestrations in endothelial cells [16,49,50]. Therefore, serum VEGF may serve as a potential biomarker for measuring treatment responsiveness and for monitoring the severity of RCD pending further longitudinal study.

Hypo-HDLemia was found to have a significant odds ratio of 2.17 when comparing patients with RCD to the control group [51]. This finding adds to the existing risk factors for RCD. In the study by Hudek et al. (2019), several fatty acids levels were found to be significantly higher in patients with RCD but the Omega-3 Index in the RCD group was significantly lower than that in the control group (5.01 ± 1.27 verses 6.01 ± 1.39, *p* = 0.03) [52]. These results align with a recent systematic review that showed an association between metabolic syndrome and RCD [8]. In Kim et al.’s (2021) study, significantly lower levels of vitamin B12, vitamin D, and phosphorus were found in serum samples of patients with RCD compared with control group patients [53]. In addition, the higher vitamin D levels were calculated to have a significantly lower odds ratio (0.89, 95% CI 08.2–0.96, *p* = 0.006) for degenerative RC tears. Summarizing the results from Savitskaya et al. (2011), Park et al. (2018), Hudek et al. (2019), and Kim et al. (2021), we propose the hypothesis that the pathological mechanism of RCD may involve systemic inflammatory components. These substances correlate with diverse metabolic factors such as body composition, age, diet, and exercise habits. Further investigation is required in order to determine the exact pathological mechanism between these substances and RCD as well as their usefulness as specific indicators for detecting or monitoring RCD.

A lower HDL level was statistically correlated with RCD (odds ratio = 0.99, *p* = 0.035) [51] but its clinical relevance remains to be determined. Similarly, higher NPN was found in patients with RCD than with those in the control group [54] but the difference (37.2 ± 8.9 mg/dL versus 35.9 ± 10.2 mg/dL, effect size = 0.14 Cohen’s d) may be too small to draw a clinical conclusion from. Although Longo et al. (2009) reported significantly higher fasting plasma glucose levels in the RCD group, values were within the normoglycemic range [55]. Furthermore, the fasting serum glucose level of patients with RCD reported in Kim et al.’s (2021) study was rather insignificant [56]. Other findings were that the levels of biomarkers MMP-1,2,3,7,9, TIMP-2,3,4, LDL, TG, non-HDL, triglycerides, cholesterol, fibrinogen, and creatinine were similar among RCD and control group patients.

Although several recent studies did not meet the inclusion criteria of this systematic review as a result of their methodologies, their results provide important insights. Hedderson et al. (2020) investigated plasma samples of patients with acute (<48 h) shoulder muscle injuries not specific to RC. In contrast to Savitskaya et al.’s (2011) study which reported significantly lower levels of serum IL-10 in patients with RCD, significantly higher levels of plasma IL-6 and IL-10 were reported in Hedderson et al.’s (2020) study [13,57]. Their results indicate that the pathology of RCD differs from that of acute inflammation. Lee et al. (2020) conducted research comparing two groups of patients, those who had revision surgery within 2 years after the first repair, and who did not require additional surgery during the same period. Their results showed significantly higher levels of serum total cholesterol (210.2 ± 40.0 versus 189.7 ± 39.1, *p* = 0.012) and low-density lipoprotein (130.7 ± 28.7 versus 115.5 ± 26.9, *p* = 0.008) in the group with revision surgery [58]. The findings are different from Park et al.’s (2018) study which reported no significant differences between the serum total cholesterol levels in patients with RCD and those in the control group (200 ± 35 versus 198 ± 37, *p* > 0.05) [51]. Therefore, even serum cholesterol level may not be specific for detecting RCD but could be an indicator for planning RCD treatments. Similarly, Suh et al. (2020) conducted a research study to evaluate serum levels of high-sensitive C-reactive protein and high-density lipoprotein in patients with RCD. However, their inclusion was specific to patients with RCD and hand osteoarthritis, and which condition the results were due to was uncertain [59].

In addition to the inflammatory markers and biochemical substances mentioned in this review, a recent study has shown the presence of estrogen and progesterone receptors in supraspinatus tendon biopsies of patients with RCD [60]. Therefore, sex hormones can also be potential markers to detect physiological changes in RCD since estrogen and testosterone have been proposed to play important roles in tendon collagen synthesis and turnover, respectively [61].

Serological assessment for biomarkers is convenient and less invasive than biopsy studies; however, techniques based on Enzyme-Linked Immunosorbent Assays (ELISA) may only be available in specialized laboratories, often at high costs. The use of biosensors such as electrochemical, optical, quartz crystal microbalance, and wearable biosensors can provide more rapid, non-invasive ways to detect nucleic acids, enzymes, antibodies, and peptides [62]. Such techniques are available for osteoarthritis studies, thus they may also be applicable for RCD investigations in addition to studies for other musculoskeletal diseases.

### 4.1. Limitations

Overall, the number of serological studies included in this review is less than the previous review on biopsy studies [5,6]. For article selection, automated filters were used; relevant articles may have been excluded if they were not properly classified. The studies included in this systematic review are all cross-sectional studies without repetitive longitudinal assessment; therefore, it remains unclear if levels of biomarkers change after treatment to reflect responsiveness. Certain risks of bias may exist based on the quality of the methodology. All included studies are level III evidence, yet the overall level of evidence is low. Meta-analysis was not conducted since the included studies do not share similar outcome measures.

In most of the included studies diagnosis of RCD was confirmed by MRI, but in the study conducted by Hallgren et al. (2012), diagnosis was verified by sonographic examination [14]. Furthermore, Papalia et al. (2011) did not specify the diagnostic imaging means used for subject inclusion in their study [54]. Some of the studies did not indicate specifically whether subjects with RC tears were symptomatic or asymptomatic; for example, the study group used by Park et al. (2018) consisted of patients with both symptomatic and asymptomatic RC tears [51]. Patients with comorbidities were excluded from this systematic review, but in actual situations patients with RCD around age 40–60 may have other comorbidities such as osteoarthritis or diabetes mellitus.

### 4.2. Future Research

As mentioned, several immune biomarkers were found to be elevated in the biopsy samples of RCD patients including COX-1, COX-2, TNFα, and HIFs, and should be considered for further serological studies. Additional longitudinal research on blood biomarkers including VEGF, ANG, IL-1β, IL-8, and IL-10 may determine their relevance as suitable and sensitive biomarkers in monitoring RCD disease status.

## 5. Conclusions

This is the first systematic review to summarize the case-control and cohort studies of serological investigations in RCD patients. The results of the included studies demonstrate significant changes in the levels of some biochemical substances in blood samples of RCD patients. The findings indicate systemic changes occurring in patients with RCD, and the review’s results provide new insights in understanding the complex pathology of RCD. Secondly, the results suggest serological tests for certain biochemical substances can be possibly applied clinically to monitor the status of RCD, which could be helpful in management plans. Further longitudinal studies are expected to evaluate further the usefulness and sensitivity of serological tests for RCD.

## Figures and Tables

**Figure 1 medicina-58-00301-f001:**
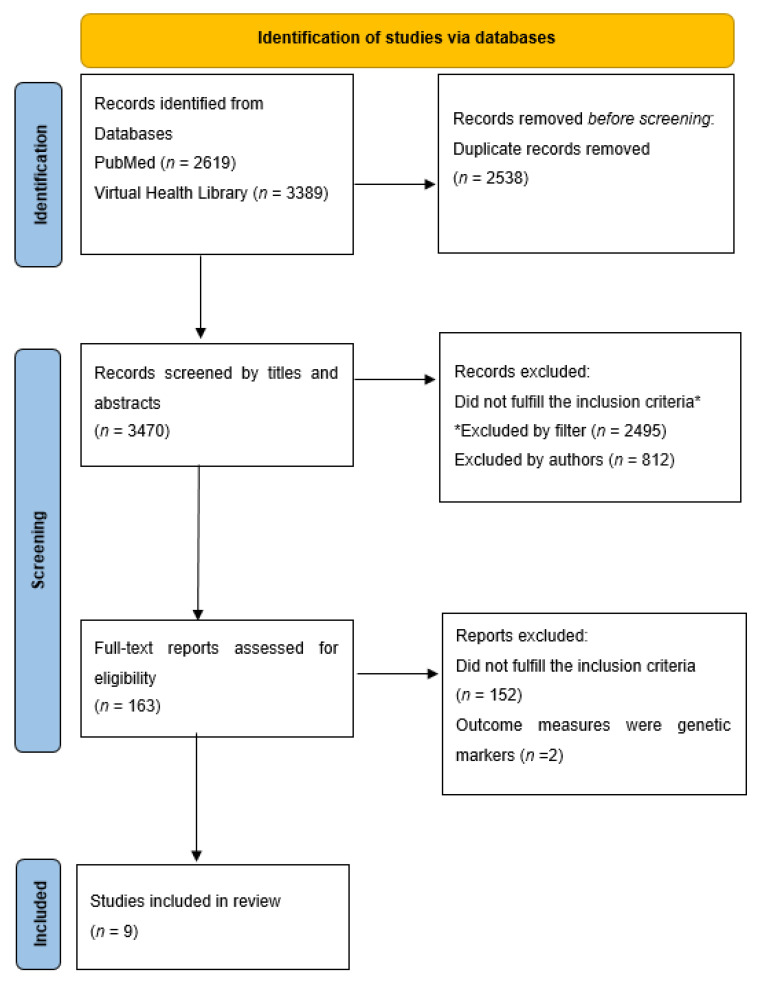
Diagnosis of RCD was confirmed by radiological or surgical examination. Exclusion criteria—study populations involving other shoulder pathologies such as fractures, frozen shoulders, and shoulder joint dislocations were excluded. Subjects with other systemic conditions or comorbidities which could affect the outcomes were excluded. In addition, the studies were accompanied by suitable control groups without pathologies in the shoulders or systemic conditions such as inflammatory joint disease, immunological, or neoplastic disorders. All animal studies were excluded. (* Filter by English language, human subjects, and study designs).

**Table 1 medicina-58-00301-t001:** Summary of study characteristics and results of included studies.

Author	Study Design and Level of Evidence	Population and Diagnosis	Biomarkers	Results
Hallgren et al., 2012	Case-Control Level III	Experimental group 17 patients with sonographically verified rotator cuff tears Mean age 61 (39–77) years Control group # 16 age and sex-matched individuals with no history of shoulder disease and sonographically intact rotator cuffs	Plasma samples MMP-1 MMP-2 MMP-3 MMP-7 MMP-9 TIMP-1 TIMP-2 TIMP-3 TIMP-4	TIMP-1, median (range) Experimental group 86 (67–119) ng/mL Control group 78 (66–93) ng/mL Measured by Luminex, *p* = 0.04 Not significant when repeated using ELISA method, *p* = 0.2 Other measures not significant
Hudek et al., 2019	Case-Control Level III	Experimental group 29 patients with non-traumatic complete rotator-cuff tears (22 males, 7 females; mean age: 53.9 ± 7.1 years) Diagnosis confirmed by MRI and surgery Control group 15 non-smoking subjects with healthy shoulders (10 males, 5 females; mean age 52.5 ± 6.5 years)	Plasma sample Fatty acids C16:1n-7 C20:2n-6 C24:0 C24:1n-9 C22:6n-3 Omega-3 Index	C16:1n-7 Exp: 0.59 ± 0.29 Control: 0.33 ± 0.12 * *p* < 0.01 C20:2n-6 Exp: 0.22 ± 0.04 Control: 0.19 ± 0.02 * *p* = 0.02 C24:0 Exp: 0.71 ± 0.24 Control: 0.52 ± 0.21 * *p* = 0.01 C24:1n-9 Exp: 0.78 ± 0.17 Control: 0.65 ± 0.20 * *p* = 0.04 C22:6n-3 Exp: 4.23 ± 1.11 Control: 5.10 ± 1.12 * *p* = 0.02 Omega-3 Index Exp: 5.01 ± 1.27 Control: 6.01 ± 1.39 * *p* = 0.03
Park et al., 2018	Cohort study Level III	Experiment group 199 patients with posterosuperior rotator cuff tears confirmed by MRI 143 patients were symptomatic 107 male (53.8%) Age 61.9 ± 7.6 Control group 435 individuals with intact RC 200 male (46.0%) Age 57.7 ± 8.8	Serum lipid levels Total cholesterol LDL TG HDL Non-HDL	Experimental group Total cholesterol 200 ± 35 LDL 133 ± 33 TG 109 (83–60) HDL 54 (45–63) Non-HDL 145 ± 35 Control group Total cholesterol 198 ± 37 LDL 131 ± 34 TG 104 (78–150) HDL 57 (46–67) Non-HDL 140 ± 37 Unit: mg/dL HDL odds ratio = 0.99 (0.98–1.00), *p* = 0.035 Other measures not significant Dyslipidemia–Hypo-HDLemia Experimental group = 62 (31.2%) Control group = 75 (17.2%) Hypo-HDLemia odds ratio = 2.17 (1.47–3.21), * *p* < 0.001
Longo et al., 2009	Case-control Level III	Experimental group 97 patients with rotator cuff tears (36 men and 61 women; mean age: 62.9 years, range 37 to 82) Diagnosis was based on clinical and imaging grounds and surgery Control group 97 patients with no evidence of shoulder pathologies (36 men and 61 women; mean age: 61.6 years, range 36 to 80)	Venous fasting plasma glucose levels	Experimental group 99.17 ± 9.04 mg per decilitre 5.5 ± 0.5 millimoles per litre Control group 95.45 ± 9.87 mg per decilitre 5.3 ± 0.55 millimoles per litre * *p* < 0.01, both groups were within the normoglycemic range
Longo et al., 2010	Case-control Level III	Experimental group 120 patients underwent arthroscopic repair of rotator cuff tears (45 men and 75 women; mean age: 64.86 years) Diagnosis was confirmed by MRI Control group 120 patients underwent knee arthroscopic meniscectomies with no evidence of shoulder pathologies (45 men and 75 women; mean age: 63.91 years)	Serum triglyceride and total cholesterol concentrations	Serum triglycerides Unit in mg per decilitre and millimoles per litre Experimental group Male 158.42 ± 122.29 1.81 ± 1.39 Female 131.81 ± 55.9 1.49 ± 0.63 Control group Male 139.87 ± 75.56 1.58 ± 0.85 Female 120.48 ± 53.75 1.36 ± 0.61 Serum cholesterol Unit in mg per decilitre and millimoles per litre Experimental group Male 212.76 ± 40.58 5.51 ± 1.05 Female 224.11 ± 44.42 5.80 ± 1.15 Control group Male 213.6 ± 36.45 5.53 ± 0.94 Female 217.3 ± 39.28 5.63 ± 1.02 not statistically significant differences either in triglyceride concentration (*p* = 0.6) or total cholesterol concentration (*p* = 0.1)
Longo et al., 2014	Case-control Level III	Experimental group 82 patients underwent arthroscopic repair of RC tears (36 men and 46 women; mean age: 57.7 ± 10.2 years) Diagnosis confirmed by MRI Control group 82 patients underwent arthroscopic meniscectomies for meniscal tears with no history of RC symptoms (36 men and 46 women; mean age: 55.9 ± 9 years)	Serum fibrinogen concentration.	Experimental group 335.9 ± 171 mg/dL (range 70–512; median 328.5) Control group 329.6 ± 205 mg/dL (range 72–607; median 322.5) *p* = 0.05
Papalia et al., 2011	Case-control Level III	Experimental group 200 patients who underwent arthroscopic repair of rotator cuff tears (93 men, 107 women; age 56.8 ±11.7 years) Diagnosis was confirmed by preoperative imaging and arthroscopy Control group 200 patients who underwent knee arthroscopies for management of meniscal tears, with or without cartilage damage (93 men and 107 women, age of 53.9 ± 12.6 years)	Plasma Nonprotein nitrogen (NPN) and creatinine levels	NPN Experimental group 37.2 ± 8.9 mg/dL 2.06 ± 0.49 mmol/L Control group 35.9 ± 10.2 mg/dL 1.98 ± 0.56 mmol/L * *p* = 0.035 Creatinine Experimental group 0.8 ± 0.19 mg/dL 0.04 ± 0.01 mmol/L Control group 0.82 ± 0.18 mg/dL 0.045 ± 0.01 mmol/L *p* = 0.66
Savitskaya et al., 2011	Case-control Level III	Experimental group 200 patients with significant inflammation, tendon degeneration, and partial or full-thickness rotator cuff tears (112 males, 88 females, age 40.3 ± 10.9) Diagnosis was confirmed by MRI Control group 200 age and sex matched healthy individuals with no medical history of rotator cuff disease (107 males, 93 females, age 43.3 ± 11.5)	Serum samples IL-1β IL-8 IL-10 VEGF ANG	Experimental group IL-1 16.17 ± 6.71–43.71 ± 8.91 IL-8 15.31 ± 0.85–27.81 ± 1.11 IL-10 3.11 ± 1.91–7.64 ± 1.11 VGEF 402.11 ± 88.11–621.24 ± 301.11 ANG 166.45 ± 44.90–89.39–40.19 Control group IL-1 3.33 ± 0.69 IL-8 9.11 ± 0.98 IL-10 9.53 ± 1.21 VEGF 339.67 ± 74.65 ANG 239.51 ± 58.4 Unit in pg/mL * IL-1β, IL-8, and VEGF levels were significantly higher in RCD patients than in controls. Serum ANG and IL-10 levels were significantly lower in RCD patients than in controls Exact *p* values not provided * Overexpression of VEGF correlated with advanced disease (r = 0.75; *p <* 0.01), average microvascular density (r = 0.68, *p <* 0.01), and visual analog score (r = 0.75, *p <* 0.01) in patients with RCD.
Kim et al., 2021	Case-control Level III	Experimental group 40 patients with degenerative RC tears Diagnosis was confirmed by MRI (23 males, 17 females, age 61.0 ± 5.3) Control group (*n* = 47) 40 patients with minor non-shoulder trauma but no RC tears or associated symptoms or clinical signs	Serum sample Glucose Magnesium Calcium Phosphorus Zinc Homocysteine Vitamin D Vitamin B12 Folate	Vitamin B12 Experimental group 528.4 ± 145.7 pg/mL Control group 627.1 ± 183.0 pg/mL * *p* = 0.007 Vitamin D Experimental group 15.7 ± 7.2 ng/mL Control group 21.6 ± 10.0 ng/mL * *p* = 0.002 Vit D odds ratio (OR) for degenerative RC tear = 0.89; 95% CI = 0.82–0.96; * *p* = 0.006 Phosphorus Experimental group 3.2 ± 0.6 mg/dL Control group 3.6 ± 0.7 mg/dL * *p* = 0.008 other parameters showed no significant relationships

# No exact data of genders and age were provided. Abbreviations: MMP = matrix metalloproteinase, TIMP = tissue inhibitor of metalloproteinases, IL = interleukin, LDL = low-density lipoprotein, TG = triglyceride, HDL = high-density lipoprotein, hypo-HDLemia = hypo-high-density lipoproteinemia, VEGF = Vascular endothelial growth factor, and ANG = angiogenin. ELISA = Enzyme-linked immunosorbent assay. * Results with statistical significance, *p* < 0.05.

**Table 2 medicina-58-00301-t002:** Critical appraisal of included studies. Study Quality Assessment Tool for Observational Cohort and Cross-sectional studies, the National Heart, Lung, and Blood Institute.

	Hallgren et al., 2012	Hudek et al., 2019	Park et al., 2018	Longo et al., 2009	Longo et al., 2010	Longo et al., 2014	Papalia et al., 2011	Savitskaya et al., 2011	Kim et al., 2021
1. Was the research question or objective in this paper clearly stated?	Y	Y	Y	Y	Y	Y	Y	Y	Y
2. Was the study population clearly specified and defined?	Y	Y	N	Y	N	N	Y	N	Y
3. Was the participation rate of eligible persons at least 50%?	Y	Y	Y	Y	Y	Y	Y	Y	Y
4. Were all the subjects selected or recruited from the same or similar populations (including the same time period)? Were inclusion and exclusion criteria for being in the study prespecified and applied uniformly to all participants?	Y	Y	Y	Y	Y	Y	Y	N	Y
5. Was a sample size justification, power description, or variance and effect estimates provided?	N	Y	N	N	N	N	Y	N	Y
6. For the analyses in this paper, were the exposure(s) of interest measured prior to the outcome(s) being measured?	N	N	N	N	N	N	N	N	N
7. Was the timeframe sufficient so that one could reasonably expect to see an association between exposure and outcome if it existed?	N	N	N	N	N	N	N	N	N
8. For exposures that can vary in amount or level, did the study examine different levels of the exposure as related to the outcome (e.g., categories of exposure, or exposure measured as continuous variable)?	NA	NA	NA	NA	NA	NA	NA	NA	NA
9. Were the exposure measures (independent variables) clearly defined, valid, reliable, and implemented consistently across all study participants?	Y	Y	Y	Y	Y	Y	Y	Y	Y
10. Was the exposure(s) assessed more than once over time?	N	N	N	N	N	N	N	N	N
11. Were the outcome measures (dependent variables) clearly defined, valid, reliable, and implemented consistently across all study participants?	Y	Y	Y	Y	Y	Y	Y	Y	Y
12. Were the outcome assessors blinded to the exposure status of participants?	NR	NR	NR	NR	NR	NR	NR	NR	NR
13. Was loss to follow-up after baseline 20% or less?	Y	Y	Y	Y	Y	Y	Y	Y	Y
14. Were key potential confounding variables measured and adjusted statistically for their impact on the relationship between exposure(s) and outcome(s)?	N	N	N	Y	N	N	Y	N	Y
Number of “yes”	7	8	6	8	6	6	9	5	9

Y, yes; N, no; CD, cannot determine; NA, not applicable; NR, not reported.

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
