# Peer review of "The Usefulness of Serological Inflammatory Markers in Patients with Rotator Cuff Disease—A Systematic Review"

_medicina, 2022, doi:10.3390/medicina58020301_

Round 1

Reviewer 1 Report

Dear Authors,

in order to underline the importance of inflammatory cascade and the role of citokines i suggest to cite at the end of the line 46 the following article:

  • Is extracorporeal shockwave therapy effective even in the treatment of partial rotator cuff tear?. Journal of biological regulators and homeostatic agents, 34(2), 709-714. Notarnicola, A., et al . (2020).

In fact in this article in the discussion section the Authors describe the importance of pro inflammation mediators in the cuff tear.

As regards the M&M in the eligibility criteria , it is necessary to detail the systemic conditions at line 86 that you consider in order to esclude the studies (neurological, immunological etc…..)

As regard the study selection it is neccesary to specify the experience and qualify of reviewers (CL,BL and SN). They are specialists, students etc…..?

As regards the results the section is well described.

As regards the discussions and conclusions are interesting and well balanced

Author Response

Response to Reviewer 1 Comments

  • in order to underline the importance of inflammatory cascade and the role of citokines i suggest to cite at the end of the line 46 the following article:
  • Is extracorporeal shockwave therapy effective even in the treatment of partial rotator cuff tear?. Journal of biological regulators and homeostatic agents, 34(2), 709-714. Notarnicola, A., et al . (2020).

In fact in this article in the discussion section the Authors describe the importance of pro inflammation mediators in the cuff tear.

Response:

We thank the reviewer’s comment and the citation has been added at the end of the line 46 as [7].

  • As regards the M&M in the eligibility criteria, it is necessary to detail the systemic conditions at line 86 that you consider in order to exclude the studies (neurological, immunological etc…..)

Response:

Thank you for the comment. More details about the systemic conditions have been added in line 88-89, “such as inflammatory joint disease, immunological or neoplastic disorders.”

  • As regard the study selection it is necessary to specify the experience and qualify of reviewers (CL,BL and SN). They are specialists, students etc…..?

Response:

Thank you for the suggestion and we have revised the text accordingly, with specific experience and qualifications of all 3 reviewers added in line 99-105, page 2.

  • As regards the results the section is well described.

As regards the discussions and conclusions are interesting and well balanced

Response:

We thank the review for the positive comments.

Reviewer 2 Report

The paper is well written and the methodology is robust. Finding new solutions to detect RCD is challenging but fundamental to manage this condition. However, I have some concerns that are reported in the file attached below

Author Response

Responses to Reviewer 2 comments

  • the title should reflect the main result of your paper

Response: We thank the reviwer for this important suggestion and title has been revised accordingly to “The usefulness of serological inflammatory markers in patients with Rotator Cuff Disease –a systematic review”

  • mesh term were not included

Response: Mesh terms were indeed applied although we failed to indicate in our initial submission. Appendix A1 and A2 have been updated accordingly. Mesh terms function is only available on PubMed but not VHL.

  • who performed the quality assessment?

Response: We thank the reviewer for this important point, with details now added in lines 115-117 accordingly. “Quality and risk of bias assessment were conducted by two reviewers (CL and BL) individually. Any inconsistent results were discussed and resolved together with the third reviewer (SN).“

  • automate filters are useful but could exclude relevant articles that are not proper classified. I suggest to add this as one limitation of the study

Response: We agreed with this comment and have revised the text accordingly, line 425-426, page 5. “In the article selection, automate filters were used so relevant articles could potentially be excluded as they were not properly classified”

  • 1 add the reasons for exclusion

Response: Exclusion criteria is now available as figure legand for Fig. 1, line 199-204.

  • 1 add the number of articles

Response: We thank the reviewer for the reminder and the text box (n=163) is now available in Fig. 1.

  • Table 1. add level of evidence

Response: Thank you for the suggestion, the level of evidence has been added in table 1 and lines 289-290.

  • Discussion

I have two more suggestions: biomarkers are very useful in different conditions. Moreover, not only proteins or IL could have an effect in RCD, but also hormones. I suggest to use this paper to discuss this point PMID: 34670550. Other recent techniques have been developed to detect biomarkers for OA. Maybe in the future they could also be used in RCD. PMID: 33561091

Response: We thank the reviewer for bringing up both into points and have amended the Discussion accordingly, lines 408-421, page 5.

  • Limitation -different methods were used to assess the RCD.

- the comorbidities of the patients were not included

- the overall level of evidence is low

-it was not possible to perform a metanalysis

Response: We thank the reviewer for the above comments and have revised Section 4.1 “Limitations”, page 5 accordingly.

  • The search string is poor. no mesh terms were used and only two databases were included. I suggest to screen other sources as scopus, embase and google scholar

Response: Thank you for the comment. We would like to clarify that the search engine Virtual Health Library (VHL) has covered the databases MEDLINE, LILACS, IBECS and BENF Nursing. Google Scholar was also used but it could not apply a systematic search list so it was not indicated in Fig.1. Clarification has been added in lines 70-76. After removing duplicate records, there were more than 3000 records from these databases. Therefore, we think the search records from these databases were sufficient.

Mesh terms were indeed applied but sorry that we did not indicate properly. Appendix A1 and A2 have been updated accordingly (Remarks: Mesh terms function is only available on PubMed but not VHL).

Round 2

Reviewer 2 Report

The authors answered all my comments